# PLUM: Improving Inference Efficiency
# By Leveraging Repetition-Sparsity Trade-Off

**Sachit Kuhar**,* **Yash Jain, Alexey Tumanov**
*Georgia Institute of Technology*

**Reviewed on OpenReview:** *https://openreview.net/forum?id=IEKtMMSblm*

## Abstract

Efficient inference of Deep Neural Networks (DNNs) on resource-constrained edge devices is essential. Quantization and sparsity are key techniques that translate to repetition and sparsity respectively within tensors at the hardware-software interface. This paper introduces the concept of repetition-sparsity trade-off that helps explain computational efficiency during inference. We propose PLUM, a unified co-design framework that integrates DNN inference systems and quantization (forward and backward pass) to leverage repetition-sparsity trade-off to improve inference efficiency. Our results demonstrate that PLUM's quantization method is more accurate than binary quantization with the same number of non-zero weights. Detailed analysis indicates that signed binarization generates a smaller distribution of effectual (non-zero) parameters nested within a larger distribution of total parameters of latent full-precision weights for a DNN block. Finally, the proposed PLUM framework achieves a 26% speedup on real hardware, doubles energy efficiency, and reduces density by $2.8\times$ compared to binary methods while retaining top-1 accuracy when compared to prior-art methods for ResNets on ImageNet (by achieving 66.2% top-1 accuracy), presenting an alternative solution for deploying efficient models in resource-limited environments. Code available at https://github.com/sachitkuhar/PLUM

## 1 Introduction

Despite significant strides in accuracy, the burgeoning complexity and resource demands of deep learning models pose challenges for their widespread adoption across a wide range of domains (He et al., 2016; Brown et al., 2020; Graves et al., 2013; Jain et al., 2022; Mandlekar et al., 2021; Jain et al., 2024). This requires the development of innovative techniques to enhance DNN efficiency during inference on edge devices. Two such techniques have been studied extensively: binarization and sparsity. Binarization, a form of quantization, results in weight repetition as only two values appear repeatedly in the weight tensor (Courbariaux et al., 2015). This approach significantly trims the memory footprint of the weight tensor, thereby decreasing memory I/O during inference (Hegde et al., 2018). In contrast, sparsity leads to zero weight values. Since anything multiplied by zero is zero, weight sparsity leads to *ineffectual* multiplications (Wu et al., 2021). This approach reduces memory I/O during inference by not reading activations that would be multiplied by zero weights (Gong et al., 2020). Thus, both these techniques are geared towards reducing memory I/O during inference.

DNN inference efficiency is usually achieved by leveraging either binarization or sparsity. Introducing sparsity by adding a zero-valued weight in conjunction with binary weights is termed ternary quantization. Ternary was conceived with the reasonable assumption that transitioning from binary to ternary models would only minimally impact inference latency due to the effect of zero weights (Li et al., 2016). However, advances in the contemporary hardware-software systems have revealed a substantial increase in latency during such transitions (Prabhakar et al., 2021; Fu et al., 2022).This work aims to perform faster inference on contemporary hardware-software systems while retaining model accuracy.

---

*Principal Investigator; Correspondence: kuhar.sachit@gmail.com

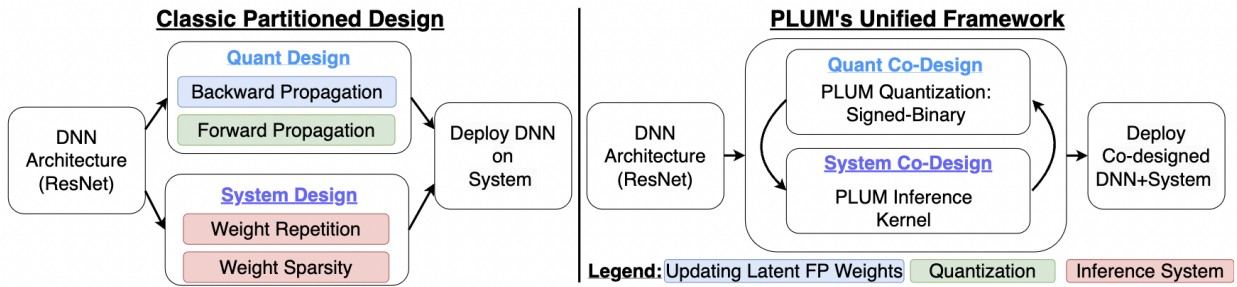

Figure 1: **On the left:** The conventional, isolated approach where DNN inference systems and quantization methods are designed separately, resulting in being ignorant of repetition-sparsity trade-off, leading to inefficient inference. **On the right:** PLUM, a unified design framework, performs quantization-system co-design to exploit the repetition-sparsity trade-off, thereby enhancing computational efficiency.

The key insight of this work is the concept of the *repetition-sparsity trade-off*. This trade-off explains the inference inefficiency of binary and ternary weight quantization. A traditional binary network chooses maximization of weight repetition while being ignorant of weight sparsity, whereas a ternary network introduces weight sparsity at the expense of weight repetition. For instance, transitioning from binary to ternary networks increases the number of possible unique 3x3 2D filters from 512 ($2^9$) to 19683 ($3^9$). This makes it exponentially harder to extract efficiency using filter repetition.

The conventional approach to deploying efficient DNNs has largely been a two-stage process: training the DNN with binarization or ternarization (Bai et al., 2018), followed by formulating a system design that reduces memory I/O by leveraging both weight repetition and/or sparsity (Hegde et al., 2018). This partitioned design approach has revealed a pitfall: increased memory I/O during inference due to the repetition-sparsity trade-off. However, recent advancements in DNN inference systems can now exploit repetition and sparsity to improve efficiency as shown in (Prabhakar et al., 2021; Fu et al., 2022). To this end, this paper proposes a quantization-system co-design framework called **PLUM** (**PLU**s-**M**inus) **Framework** as seen in Figure 1.

This framework aims to demonstrate and leverage the existence of repetition-sparsity trade-off. PLUM re-imagines different levels of stack working together in synergy to enhance computational efficiency during inference while retaining the model's accuracy. As seen in Figure 2, compared to a classic partitioned design for binary, the PLUM framework improves inference efficiency (with respect to energy, latency, and throughput) and reduces model density to allow the use of bigger models, resulting in more accuracy. Based on our analysis and empirical evidence, we envision that the PLUM framework has the potential to positively shape the evolution of more efficient and high-performing deep learning models.

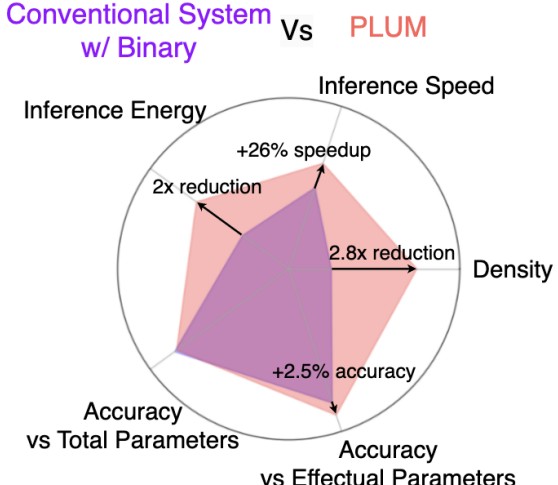

Figure 2: **PLUM vs. Prior-Art**: For ResNet on ImageNet, the pronounced spread indicates that PLUM holistically outperforms prior-art method of partitioned design using binary quantization. It retains competitive accuracy and pushes the Pareto front, exhibiting a +2.5% improvement when both methods employ a comparable number of effectual parameters. Moreover, our method enhances inference efficiency, achieving a 26% speedup, doubling energy efficiency, and reducing density by 2.8x for the same backbone.

We make the following contributions in our work:

- We present the concept of repetition-sparsity trade-off along with the PLUM framework. While the former explains the inference inefficiency of binary and ternary weight quantization, the latter leverages (and thus proves) this insight to improve the efficiency of DNN inference.

- Compared to prior-art quantization schemes, we demonstrate that PLUM's signed-binary quantization scheme retains model accuracy (of 90.7% and 66.2% top-1 accuracy on CIFAR10 and ImageNet) while requiring significantly fewer effectual parameters. We offer detailed insights on co-dependent learned features in the DNN when using PLUM by visualizing the distribution of quantized and latent full-precision weights.

- We perform DNN inference on Intel CPUs to prove the repetition-sparsity trade-off and demonstrate that the PLUM framework leads to the fastest inference. We further demonstrate the benefits of sparsity during inference when using PLUM via energy reduction experiment.

## 2 Background

**Quantization**  Quantization in deep learning involves mapping of real-valued numbers to a select set of discrete values to streamline and optimize computations. The Binary Quantization technique assigns any real-valued number to either +1 or -1 (Courbariaux et al., 2015). The primary objective behind this is to simplify operations in DL hardware, turning multiply-and-accumulates (MACs) into mere accumulate operations (Rastegari et al., 2016). Conversely, Ternary Quantization designates values to +1, -1, or 0 (Li et al., 2016). This determination is based on a threshold, $\Delta$. Values exceeding $\Delta$ are assigned +1, those below receive -1, while others become zero. For both quantization methods, the output values can undergo scaling using a factor, $\alpha$ (Bulat & Tzimiropoulos, 2019; Bulat et al., 2019). There is a rich literature on improving the accuracy of these methods (Qin et al., 2023; Zhang et al., 2018; Gong et al., 2019; Qin et al., 2020b; Hu et al., 2018; Pouransari et al., 2020; Liu et al., 2018). These methods are not aware of the repetition-sparsity trade-off. In PLUM framework, we combine the benefits of both binary and ternary to create efficient inference.

**Weight Sparsity**  Weight Sparsity in DNN implies that there is repetition of weight with a value equal to zero. The basic idea is that since $0 \times x = 0$ for any real-valued scalar $x$, if the weight is zero, the multiplication is *ineffectual* and should be skipped (Gong et al., 2020). Sparsity in weights is static during inference (Dai et al., 2020). Therefore, if the value of weight is zero, we can choose not to load activations corresponding to that weight (Qin et al., 2020a). This can lead to a reduction in data movement, memory accesses, and MACs thereby reducing computations and hence resulting in efficient DNN inference. This approach has been effective on ASICs and general-purpose devices (Hegde et al., 2019; Wang et al., 2021; Dai et al., 2020; Gong et al., 2020; Nayak et al., 2023). PLUM exploits weight sparsity during inference to improve efficiency.

**Weight Repetition**  Quantization of weights leads to the same value being repeated again and again in the weight tensor. This phenomenon is known as weight repetition (Hegde et al., 2018; Sze et al., 2020). Since the weights are fixed during DNN inference (Dai et al., 2020), this leads to opportunities for improving efficiency during inference with respect to time and energy by exploiting the repetition of weights and reducing memory accesses (Sze et al., 2020). This concept was first introduced in BNN (Courbariaux et al., 2016). They demonstrated that for a CNN, only 42% of filters are unique per layer on average which can lead to reducing the number of operations by 3x.

UCNN (Hegde et al., 2018) demonstrated how to leverage widespread and abundant weight repetition present across a range of networks (like ResNet, GoogleNet) that are trained on a variety of datasets. It reorders the weights and thus reorders activations, leading to reduced memory access and arithmetic operations, resulting in 4x more efficient inference. For example, if the filter weights are $[a, b, a, a]$ and activations are $[w, x, y, z]$, UCNN would reorder it as $a \times (w + y + z) + b \times (x)$ for efficient inference (Sze et al., 2020) but does not exploit weight sparsity. SumMerge (Prabhakar et al., 2021) uses both weight repetition and weight sparsity for efficient DNN inference. For example, if $b = 0$ in the previous example, SumMerge would

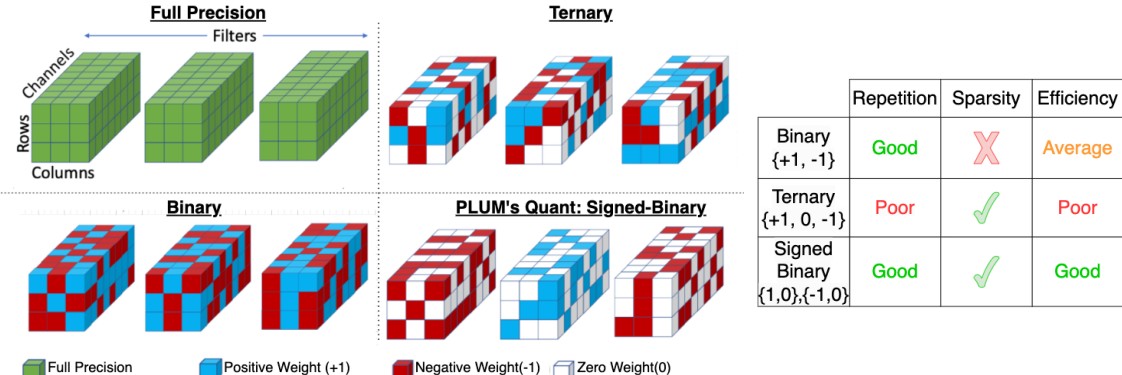

Figure 3: **Concept Diagram on the left:** The diagram shows the comparison of Binary, Ternary, and Signed Binary Quantization in terms of visual representation of their quantized weights. **Qualitative Evaluation on the right:** The table qualitatively evaluates them in terms of weight sparsity, weight repetition, and inference efficiency.

compute $a \times (w + y + z)$ during inference. While it reduces arithmetic operations up to 20x, it uses a greedy strategy to reduce arithmetic operations. Q-Gym (Fu et al., 2022) treats weight repetition as a combinatorial optimization problem to perform inference on CPUs and GPUs. But, all these methods are designed for standard pre-built quantization and assume repetition and sparsity in quantized DNNs to be rigid. All these systems do not perform the dot product of activation and weight tensors in a single step. Instead, they split the dot products of input and weights into smaller blocks via tiling to improve data locality during inference. PLUM exploits this for faster inference through quantization and DNN-inference co-design.

**Repetition, Sparsity & Inference Latency** Weight repetition is prevalent across a range of DNNs trained on diverse datasets (Hegde et al., 2018). The impact of weight sparsity on weight repetition can be most easily visualized for $3 \times 3$ convolution filters as shown in Figure 3. Binary models have $2^9$ unique filters while ternary models employ $3^9$ unique filters. Recent works (Hegde et al., 2018; Prabhakar et al., 2021; Fu et al., 2022) show a significant slowdown in inference when using ternary when compared with binary. This extends to bit-utilization; binary systems can potentially be represented by a single bit, whereas ternary systems require two bits due to the inclusion of zero-valued weights. Moreover, recent findings indicate that runtime doubles when transitioning from one to two bits during the deployment of ResNet18 on ARM CPUs, showcasing the influence of bit-width distinction on inference time (Cowan et al., 2020).PLUM aims to balance out exploitation of repetition and sparsity for faster inference.

## 3 PLUM

The objective is to create a framework that leverages trade-off between repetition and sparsity. It aspires to retain competitive accuracy while combining the merits of both repetition and sparsity for faster inference.

### 3.1 Leveraging Repetition-Sparsity Trade-off

Conventional binary networks offer 2 unique weight choices per element, creating a dense setup. Conversely, ternary networks take an additional bit to represent 3 unique weight choices per element to create a sparse setup. Continuing the $3 \times 3$ convolution filter illustration from the previous section, binary models can have $2^9$ unique filters. In contrast, ternary models employ $3^9$ unique filters, inadvertently causing an exponential decrease in repetition by a factor of $38.4\times$. These design decisions can be interpreted as follows: Binary quantization prioritizing the maximization of weight repetition, overlooking the potential benefits of weight sparsity. In contrast, ternary quantization induces weight sparsity but compromises weight repetition. This disparity places the two schemes at opposite ends of a spectrum, creating a *repetition-sparsity trade-off* (as

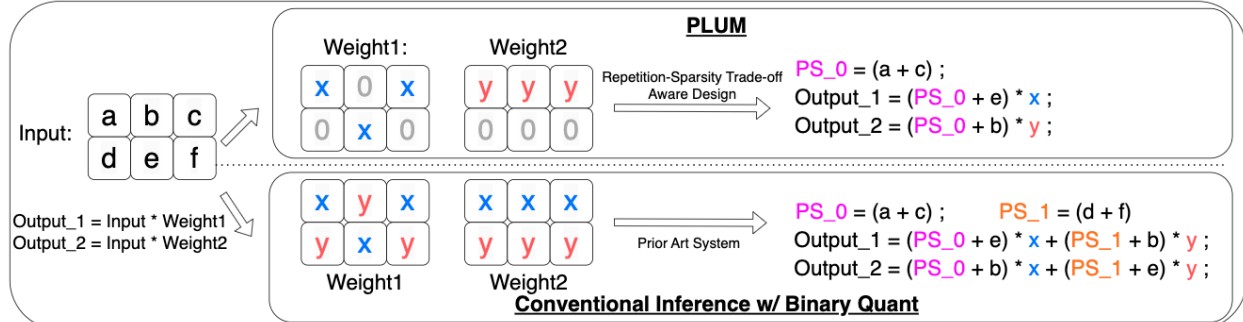

Figure 4: **PLUM framework leads to efficient inference as it acknowledges repetition-sparsity trade-off through co-design**: Visualizing inference when using recent systems (Prabhakar et al., 2021; Fu et al., 2022). Weight repetition enables binary models to skip work by re-using partial sums within and across filters. PLUM takes this even further by reducing the number of effectual operations by leveraging sparsity while retaining repetition. (details in Supp B)

seen in Figure 3). In a given DNN block, we have the flexibility to assign each latent full-precision weight one of four possible quantization function value sets: {1,-1}, {1,0}, {0,-1}, and {1,0,-1} This variety of assignments to individual weights allows for the creation of unique configurations within the block, each representing a distinct point along the repetition-sparsity spectrum. PLUM framework aims to identify a more efficient point on the repetition-sparsity spectrum. PLUM's quantization, i.e., signed-binary makes the design decision to use two quantization functions with value sets {1,0} and {0,-1} as they are sparse and could potentially be represented using one bit per latent full-precision weight.

Let the convolutional layer have a $R \times S$ kernel size with $C$ input channels and $K$ filters. The quantization function takes latent full-precision weights $W$ as input and outputs the quantized weight $W^{quant}$. The quantized weight $W^{quant}$ would be the product of the sign-factor $\beta$ and the bitmap $U$.

$$Q : W \rightarrow W^{quant}; \quad W^{quant} = \beta U \tag{1}$$

$$\forall W \in \mathbb{R}; \quad \beta \in \{+1, -1\}; \quad U \in \{0, 1\} \tag{2}$$

## 3.2 Co-design Exploration

Developing an efficient framework requires a redesign due to its distinct nature from existing methods, as a latent FP weight can be assigned at any of the two quantization functions. This leads to challenges: (1) retaining the ability to learn meaningful representations and (2) providing faster inference when using state-of-the-art inference systems. Our method addresses these concerns by revisiting design decisions on different layers of the stack with the goal of improving efficiency while retaining accuracy. For in-depth insights into our experimental setup and implementation details, please refer to the supplementary section C. For additional ablations, please see supplementary E F and G.

### 3.2.1 DNN inference in PLUM

The aim is to achieve faster inference by using two quantization functions for a DNN block. As explained in Section 2 (weight repetition paragraph), modern DNN inference systems enhance data locality during inference by splitting the dot products of inputs and weights into smaller blocks through tiling. For instance, Prabhakar et al. (2021) divides every filter into smaller blocks by introducing a sub-dimension denoted as $C^*$, representing the tile size along the $C$ dimension. We leverage this insight and make the design decision of *local binarization*. This allows a single processing step during PLUM inference to see one signed binary quantization function, leading to efficiency as shown in Figure 4. We define the region to be sign-binarized as $R \times S \times C_t$ where $C_t = \max(C, kC^*)$ and $k \in \mathbb{Z}^+$. For a given $\beta$, $\mathbf{W}^{quant} = \beta \mathbf{U}$ where $\mathbf{W} \in \mathbb{R}^{R \times S \times C_t}$ and $\mathbf{U} \in \{0, 1\}^{R \times S \times C_t}$. Hence, this results in *global ternarization* for a DNN block. Please refer to section 5.1 for ablation on DNN inference for PLUM on Intel CPU.

| Arch | FP | T | B | SB |
|---|---|---|---|---|
| ResNet20 | 92.10 | 90.86 | **90.20** | 90.05 |
| ResNet32 | 92.90 | 92.03 | 91.51 | **91.55** |
| ResNet44 | 93.30 | 92.40 | 91.93 | **91.98** |
| ResNet56 | 93.63 | 92.90 | 92.42 | **92.52** |
| ResNet110 | 93.83 | 93.33 | 92.64 | **92.68** |

| %$\{0,1\}$ filters | %$\{0,-1\}$ filters | Acc |
|---|---|---|
| 0 | 1 | 88.84 |
| 0.25 | 0.75 | 89.32 |
| 0.5 | 0.5 | **90.05** |
| 0.75 | 0.25 | 89.30 |
| 1 | 0 | 89.07 |

Table 1: PLUM's Signed-Binary does not lead to accuracy degradation wrt Binary while leading to efficient inference.

Table 2: Equal percentage of usage of %$\{0,1\}$ and %$\{0,-1\}$ regions leads to best model accuracy in PLUM.

| EDE† | Acc |
|---|---|
| Disabled | 88.4 |
| Enabled | **88.7** |

| Region | Acc |
|---|---|
| $C_t = C$ | **88.6** |
| $C_t = C/2$ | 87.9 |

| $\Delta$ | Acc |
|---|---|
| $0.01 \times \max|\mathbf{W}|$ | 90.01 |
| $0.05 \times \max|\mathbf{W}|$ | **90.05** |

Table 3: Enabling Adapted EDE† during backpropagation improves model accuracy in PLUM.

Table 4: Region size of $C_t = C$ (wrt $R \times S \times C_t$) leads to competitive accuracy in PLUM.

Table 5: Quantization functions used in PLUM are not sensitive to the choice of threshold $\Delta$.

**Ablations for PLUM**: ResNet models trained on CIFAR-10 dataset for the method. Tables 2-5 use ResNet20. Default configurations for our method are marked in grey . † Adapted Error Decay Estimator (EDE) Qin et al. (2020b) details in Section 3.2.3.

### 3.2.2 Quantization in PLUM: Signed Binary

The challenges arise from the need to design signed-binary quantization functions, that enable the model to learn meaningful representations. Local binarization techniques must accommodate a global ternarization strategy, which presents additional challenges such as determining suitable regions for local binarization and assigning these regions to their designated signed binary quantization functions. In response to these challenges, we have designed our method, supported by systematic ablation studies conducted on ResNets trained with the CIFAR-10 dataset (see supplementary I for additional ablations).

**Signed Binary Quant Functions**: Our method involves the strategic use of two distinct quantization functions (as shown in Figure 3). In this section, we intend to meticulously design these functions, characterized by the value sets $\{0,1\}$ where the sign-factor $\beta_1$ is equal to 1, and $\{0,-1\}$ where the sign-factor $\beta_{-1}$ is -1. The scaling factor $\alpha_i$ mirrors $\beta_i$ where $i = \pm 1$. Following (Zhu et al., 2016), we define the threshold value as $\Delta = 0.05 \times \max(|\mathbf{W}|)$. To assess the efficacy and sensitivity of the chosen thresholds, we experiment with different $\Delta$s using signed-binary quantization (see Section 4.2.2) to find stable performance across various configurations, as depicted in Table 5. These are defined as:

$$\mathbf{W}^{quant} = \left\{ \begin{array}{ll} \alpha_1 & \text{if } \mathbf{W} \geq \Delta \text{ and } \beta = 1 \\ \alpha_{-1} & \text{if } \mathbf{W} \leq -\Delta \text{ and } \beta = -1 \\ 0 & \text{Otherwise} \end{array} \right\} \tag{3}$$

$$\frac{\partial L}{\partial \mathbf{W}} = \left\{ \begin{array}{ll} \alpha_1 \times \frac{\partial L}{\partial \mathbf{W}^{quant}} & \text{if } \mathbf{W} > \Delta \text{ and } \beta = 1 \\ -\alpha_{-1} \times \frac{\partial L}{\partial \mathbf{W}^{quant}} & \text{if } \mathbf{W} < -\Delta \text{ and } \beta = -1 \\ 1 \times \frac{\partial L}{\partial \mathbf{W}^{quant}} & \text{Otherwise} \end{array} \right\} \tag{4}$$

**Intra-Filter Signed-Binary Quant** In this section, we delve deeper into signed binary quantization by introducing variations in $C_t$ with respect to the constant value $C$, aiming to identify the optimal setting for $C_t$. We design this approach to emphasize the changes in performance across different thresholds. Table 4 assesses the impact on representation quality by adjusting the $C_t$ values during the training. The result shows that intra-filter signed binary co-design preserves a competitive level of representation quality even with a reduced $C_t$ and setting of $C_t = C$ works best.

**Inter-Filter Signed-Binary Quant** Building on intra-filter, where $C_t = C$, this is the simplest signed binary quantization, and we term it as Inter-Filter Signed binary quantization. In essence, this method involves assigning each filter to one of two distinct signed binary quantization functions, enabling an efficient representation as $C_t = C$. We contrast it with binary and ternary quantization schemes in an apples-to-apples comparison. Table 1 illustrates ResNets of varying depths trained using different quantization schemes. The signed-binary quantization maintains comparable accuracy against traditional quantization approaches across different architectures. The largest chunk of a convolution filter that can processed during PLUM inference at a given instance is the entire convolution filter itself *i.e.*, $C_t = C$. Inter-filter signed-binary quantization automatically results in the PLUM inference processing a region corresponding to one quantization function at any given time.

**Value Assignment of Signed-Binary Quant Functions** To fine-tune our network's performance, examining how different value assignments within filters affect accuracy is crucial. This experiment focuses on studying the effects of varying the proportions of filters with {0,1} and {0,-1} value assignments. As illustrated in Table 2, we explore the delicate balance between positive and negative binary values to find the best way to optimize network performance. The data indicates that having an equal mix of these values is more beneficial, clearly outperforming the methods that use only a single type of sparse quantized function. This observation underscores the importance of using both positive and negative binary regions to achieve effective learning of representations.

### 3.2.3   Backpropagation in PLUM

In this section, we delve into tuning latent full-precision weights during training to enhance model accuracy. Traditional binary networks experience fluctuations in latent weight assignments at zero due to the use of the one quantization function, i.e., sign function—manifesting as a peak of latent full-precision weights at zero (Bai et al., 2018; Qin et al., 2020b). This issue is exacerbated in PLUM. Employing two quantization functions such that individual signed binary regions co-dependently learn diverse features, results in two distinct peaks at non-zero values of $\pm\Delta$ as illustrated in Figure 6b. This dual peak structure necessitates a process to ensure stable and accurate weight updates. In response, we have adapted traditional binary's EDE (Qin et al., 2020b) that approximates the derivative of the sign function during backward propagation to improve gradient accuracy and update capability. PLUM stabilizes fluctuations in latent full-precision weights around $\Delta = \pm 0.05 \max(\mathbf{W})$ with EDE, thereby fostering improved representations as delineated in Table 3.This is done by first determining $t$ and $k$ using the equations $t = T_{\min}10^{\frac{i}{N} \times \log \frac{T_{\max}}{T_{\min}}}$, $k = \max\left(\frac{1}{t}, 1\right)$ where $i$ is the current epoch and $N$ is the total number of epochs, $T_{\min} = 10^{-1}$ and $T_{\max} = 10^{1}$. Subsequently, we adapt $g'(\cdot)$ for PLUM: $g'(x) = kt\left(1 - \tanh^2(t(x \pm \Delta))\right)$, to compute the gradients with respect to $w$ as $\frac{\partial L}{\partial \mathbf{w}} = \frac{\partial L}{\partial Q_w(\mathbf{w})}g'(\mathbf{w})$

## 4   PLUM Pushes the Pareto-Frontier

In this section, we evaluate the efficacy of PLUM in preserving the quality of learned representations when transitioning from the conventional binarization technique. Initially, we train ResNets on the CIFAR10 and ImageNet datasets using PLUM, and benchmark the results against existing *state-of-the-art* methods in subsection 4.1. Additionally, subsection 4.3 examines the latent full-precision and quantized weights in detail to provide a holistic understanding of the retained representational capacity of the trained model.

### 4.1   Comparison with other works

To ascertain the true potential of PLUM codesign, we compare PLUM's signed-binary quantization with SOTA binary-quantization methods.. We benchmark on CIFAR10 and ImageNet datasets by training ResNet-{20,32} and ResNet-{18,34} models. Conversely, model parameters are of two types, effectual (or non-zero valued) parameters and ineffectual (or zero valued) parameters (Wu et al., 2021; Nayak et al., 2023). Since only effectual parameters lead to computation during inference, we compare different binary methods on two axes: model accuracy on the Y-axis and effectual binary model parameters on the X-axis (refer to supplementary D & C for baselines and setup). As shown in Figure 5, PLUM exhibits Pareto optimality

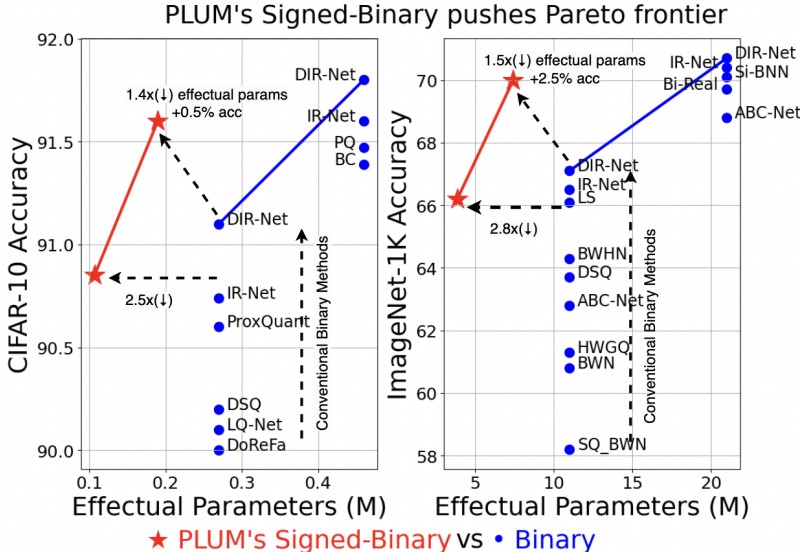

Figure 5: Comparison of PLUM and conventional binary methods on CIFAR10 and ImageNet datasets. PLUM pushes the Pareto frontier, providing superior accuracy with respect to effectual parameters and exhibiting a significant reduction in effectual parameters of equivalent models.

against state-of-the-art methods. It achieves a +0.7% and +2.5% increase in accuracy on the CIFAR10 and ImageNet datasets, respectively, coupled with 1.4× and 1.5× reduction in effectual parameters respectively. Concurrently, it realizes approximately a ∼ 2.5× and ∼ 2.8× decrease in effectual parameters for equivalent models on the respective datasets. We find no significant training overhead for PLUM when compared to conventional binary quantization schemes. Supplementary E trains signed-binary with binary till saturation under an identical setting with an equal training time of 320 epochs (4 hour training NVIDIA T4 GPU) on the CIFAR10 dataset to observe comparable accuracy with equal total parameters and higher accuracy with comparable effectual parameters. All hyperparameters are listed in supplementary D. These results emphasize the potential of PLUM co-design in creating efficient and accurate models.

## 4.2 Comparing Against Full Precision Models on Additional Datasets

We train VGG, AlexNet, and ResNet on CIFAR10, SVHN (svh, 2011), and TinyImageNet (Le & Yang) datasets, respectively. AlexNet* (ale, 2016) and VGG** (Cai et al., 2017) are derivative architectures based on AlexNet (Krizhevsky et al., 2012) and VGG (Simonyan & Zisserman, 2015) for smaller datasets as used in prior works (Prabhakar et al., 2021; Fu et al., 2022).

| Model | Dataset | Accuracy Signed Binary | Accuracy Full Precision |
|---|---|---|---|
| AlexNet* | SVHN | 97.2 | 97.7 |
| VGG** | CIFAR10 | 92.9 | 93.8 |
| ResNet18 | CIFAR100 | 75.83 | 77.8 |
| ResNet18 | TinyImageNet | 56.9 | 59.72 |

Table 6: Accuracy Comparison between Signed Binary and Full Precision Models

## 4.3 Visualizing Latent FP and Quantized Weights

**Latent FP weights.** Binarization quantization leads to a zero-mean Laplacian distribution of latent full-precision weights (Xu et al., 2021). For ResNet18 trained on Imagenet in Figure 6, we observe that despite the

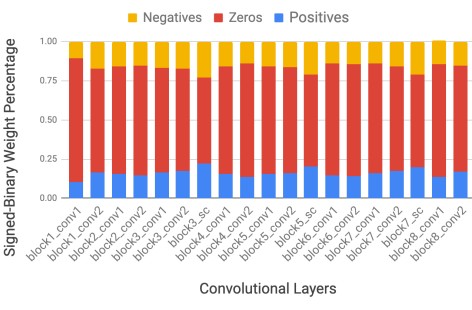

(a) **Quantized Weights**

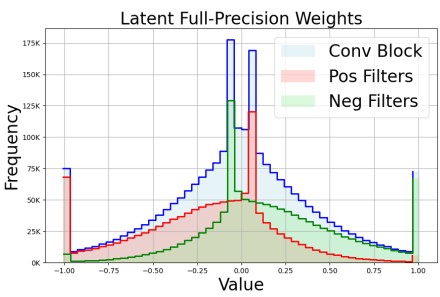

(b) **Latent Full Precision Weights**

Figure 6: **On the left: Quantized Weights**, are depicted, illustrating a distribution reminiscent of ternary networks but with weights distinctively segregated across filters, enhancing weight-repetition and inference efficiency. **On the right: Latent Full Precision Weights** in a signed-binary conv block maintain a *blue distribution* akin to binary's Laplacian. In sign-binary, *total parameters* can be divided between *positive* and *negative* signed-binary filters. These parameters can be subdivided into non-zero valued effectual and zero-valued ineffectual ones. Notably, while the total parameter's looks like binary's zero-mean Laplace distribution, individual filters do not. The effectual and ineffectual parameters' *green-red* and *red-green* distributions *resemble* Laplace and Gaussian distributions, respectively. Sign Binary reduces computations during inference when compared to binary, as it reduces effectual parameters from *blue* to *green-red* distribution.

pronounced sparsity introduced by signed binary quantization, the distribution of latent FP weights across an entire convolutional block resembles a zero-mean Laplacian distribution. Signed-binary filters are neither zero mean, nor exhibit a Laplacian distribution (supplementary H). This shows that even if individual filters don't align with a zero-mean or Laplacian distribution, collectively, the convolution block resembles these characteristics. Moreover, the presence of four distinctive peaks overlaying Laplacian resembling distribution for full-precision latent weights are due to: (a) Two peaks at the extremes appear because of clamped weights at +1 and -1, and (b) two intermediate peaks arising from the employment of threshold functions defined at $\pm\Delta$, analogous to zero-valued peak observed in binary network due to the use of sign quant function.

**Quantized weights.** We investigate the distribution in quantized weights of a trained signed-binary model by plotting the percentage distribution of quantized signed-binary convolutional blocks in ResNet18 trained on Imagenet. Figure 6 reveals a roughly consistent, equal proportion of both positive and negative weights. This observation of a roughly equal proportion of positive and negative weights is also observed in ternary quantization (Zhu et al., 2016). However, a crucial distinction arises in the distribution of positive and negative weights within a convolutional layer. Like ternary, both positive and negative valued weights are present within a layer. However, signed binary co-design *bifurcates* them across different filters. This design decision improves inference efficiency.

## 5 PLUM improves Inference Efficiency

This section shows that transitioning from conventional partitioned design for binary to PLUM co-design framework enhances inference efficiency for ResNet18 trained on ImageNet. We highlight PLUM's superiority over conventional partitioned design using binary and ternary by considering the repetition-sparsity trade-off. Additionally, we explore how weight sparsity in DNNs, specifically in signed-binary ResNet18 trained on ImageNet, benefits DNN inference.

### 5.1 Exploiting Repetition & Sparsity with PLUM

We posit that the performance of binary and ternary weight quantization methods in inference may be hindered due to their neglect of the crucial repetition-sparsity trade-off. Weight sparsity allows us to utilize upcoming sparse tensor operation technology to accelerate inference by skipping zeros during inference (Wu

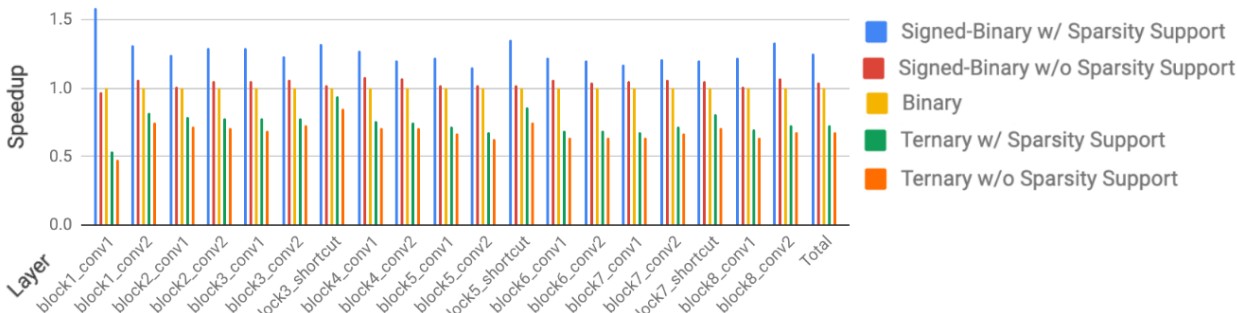

Figure 7: **Efficiency Analysis wrt Binary ResNet18 on Intel CPU**: Our study shows signed-binary excelling in every convolutional layer, depicted by bars for **signed-binary**: PLUM (w/ sparsity support) and w/o sparsity support ; **ternary**: w/ sparsity support and w/o sparsity support ; and one for **binary** , indicating its 100% density. Please refer to Figure 6. The inference computations for signed-binary w/o sparsity support is aligned with *blue* distribution, matching binary performance. On the other hand, PLUM (w/ sparsity support) reduces computation to *green-red* distribution. Because *negative* and *positive* valued quantized weights exist in separate regions of the network by design (see Figure 3), PLUM retains repetition, resulting in speedup.

et al., 2021). Concurrently, exploiting weight repetition leads to diminishing arithmetic operations and data movement (Hegde et al., 2018). However, traditional binary networks prioritize the maximization of weight repetition, overlooking the potential benefits of weight sparsity, while ternary networks induce weight sparsity but compromise on weight repetition. In contrast, we hypothesize that PLUM framework, being attuned to this trade-off, promises enhanced efficiency in actual device inference. To validate our hypothesis, we deploy quantized ResNet-18 models on Intel CPUs, rigorously measuring inference times under varying conditions.

**Experimental Setup and Methodology** We use SumMerge (Prabhakar et al., 2021) for performing inference of quantized and sparse DNNs on Intel CPUs (details in supplementary A). All experiments are conducted under identical test environments and methodologies. Within our experiments utilizing Sum-Merge, we explore two distinct configurations: (1) with sparsity support deactivated, the software does not distinguish between zero and non-zero weights, relying solely on weight repetition; (2) with sparsity support activated, the software additionally omits computations involving zero weights. We provide detailed analysis into both per-layer speedups and the aggregate speedup across different quantization strategies relative to binary quantization.

**Result and Analysis** Please refer to Figures 4 and 7. We observe in Figure 7 that PLUM, i.e., signed-binary with kernel exploiting repetition and sparsity, is most efficient for every quantized layer and the model overall by 1.26x and 1.75x faster respectively when exploiting both repetition and sparsity.

This result can be explained as follows: *(A) When sparsity support is turned off*: The software is only relying on repeating values within the weight tensor for speedup. Because binary and signed-binary have two unique values per convolutional filter, they take similar time for DNN inference. Ternary is much slower as it has three unique values per convolution filter which makes extracting efficiency by using weight repetition exponentially harder. *(B) When sparsity support is turned on*: The software not only cares about the repeating values in the weight tensor but also skips computations on zero weights to improve the runtime. Here we observe that ternary is slower than binary, because the reduction in work due to sparsity is not able to compensate for the exponential decrease in weight repetition. On the other hand, our method, PLUM, does not suffer from this problem and can exploit weight repetition and weight sparsity to the fullest and is most efficient.Thus, PLUM framework that relies on repetition-sparsity aware-inference performs better than using prior-art inference with binary and ternary methods.

**Arithmetic Operations** We report the reduction in arithmetic operations Prabhakar et al. (2021); Fu et al. (2022) required for a single inference relative to binary. We find that ResNet 18 trained using signed-binary takes 20% fewer operations during w/ sparsity support enabled inference wrt binary. On the other hand, when trained using ternary, inference requires 35% more operations during w/ sparsity support enabled inference wrt binary. Please refer to supplementary G for detailed ablations.

## 5.2 Understanding benefits of Sparsity during inference

In this section, we aim to understand the impact of weight sparsity given a fixed weight repetition in PLUM framework. We change the percentage of weight sparsity when using one bit quantization. To do this, we count the number of quantized weights with zero values and divide it by the total number of quantized weights to calculate the percentage of sparsity. We find that signed-binary ResNet-18 trained on ImageNet has 65% sparsity. Since density is (1 - sparsity) (Wu et al., 2021), ResNet-18 has 35% density (see Figure 6a). if we switch from conventional binary to PLUM's signed-binary, we decrease the density from 100% to 35%. We would like to leverage the low density to reduce the amount of computation activity during inference:

**Throughput** In a model with low density, represented by $1/x$, there is one effectual multiplication for every $x$ total multiplications. By eliminating the ineffectual computations and the time associated with them, there is the potential to improve throughput by a factor of $x$ (Emer et al., 2021). Given that the PLUM's signed-binary has a 35% aggregate density—meaning only 35% of the multiplications are effectual—exploiting sparsity during inference can lead to a potential $2.86\times$ increase in throughput compared to dense binary $\{1,-1\}$. In practice, the speedup we observe on real hardware is in the range 1.26x-1.75x as the support for unstructured sparsity on general-purpose devices is an active area of research (Wu et al., 2021). This speedup is comparable to the speedup due to unstructured sparsity on general-purpose devices in recent papers (Gong et al., 2020; Hegde et al., 2019).

**Energy Reduction** To estimate energy reduction due to the unstructured sparsity during inference, we use the cycle-level micro-architectural simulator (Muñoz-Matrínez et al., 2021) of a sparsity supporting ASIC (Qin et al., 2020a). We take their publicly released code and use it under default configuration (details in supplementary A). We observe that decreasing density from 100% to 35% for one-bit ResNet 18 leads to a ~2x reduction in energy during inference. Thus, switching from binary to PLUM's signed-binary would lead to significant improvements in power consumption on ASICs.

## 6 Discussion and Future Work

The paper introduces the concept of repetition-sparsity trade-off and proposes PLUM (PLUs-Minus) a quant-system co-design framework that leads to improvement of inference efficiency while retaining accuracy.. PLUM exhibit Pareto optimality compared to prior conventional methods, improving accuracy per effectual parameter and enhancing computational efficiency during inference with respect to speed, energy, and density. However, PLUM requires training from scratch, which can be time-consuming and computationally expensive. PLUM requires the use of two quantization functions, i.e., with value sets of $\{1,0\}$ and $\{-1,0\}$. Only using a single quantization function (just $\{1,0\}$) could lead to improving inference efficiency. Following the terminology from Section 3.1, while binary can be represented using at least $R \times S \times C \times K$ bits, signed-binary requires one additional bit per filter to signify the corresponding signed-binary quantization function, resulting in $R \times S \times C \times K + K$ bits. Further, the tile size of the modern inference system should be set such that a single processing step in PLUM should see only one signed binary quantization function. Additionally, the understudied repercussions of quantization and, consequently, signed binary quantization on model bias warrant further exploration. Despite these limitations, PLUM presents a promising approach for training efficient models that are both accurate and computationally efficient, particularly well-suited for applications where hardware resources are limited, such as mobile devices and embedded systems.

## 7 Reproducibility Statement

We provide all the hyperparameters for the key experiments including instructions on how to train the models in the supplementary C. Further, since our work is about quantization system co-design, we provide all the hyperparameters and configurations to reproduce our inference experiments in supplementary A. Finally, implementation details and baselines for our method can be found in supplementary C.

*Acknowledgements.* We thank Tushar Krishna, Christopher Fletcher, Twinkle Kuhar and Paramvir Singh for their insightful discussions, as well as the anonymous reviewers for their very helpful feedback. We'd also like to thank Francisco Muñoz-Martínez and Raveesh Garg for helping with Stonne for our experiments.

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

## A    Experiment Setup Efficiency

**Deploying on CPUs**   We use SumMerge Prabhakar et al. (2021) for this task. We run all experiments on Intel Xeon Gold 6226 CPU. In order to make our test environment as close as possible to the test environment of the authors of Prabhakar et al. (2021), we disable simultaneous multi-threading, and enable 2MB huge pages and disable dynamic frequency scaling as well. The test methodology is exactly the same as used by the authors of Prabhakar et al. (2021), i.e., each experiment is run 50 times when the machine is unloaded and the values for run with the lowest execution time are reported. All arithmetic operations are in floating-point. All DNN inference experiments are subject to identical test environments and methodology.

**ASIC**   We use STONNE Muñoz-Matrínez et al. (2021), a cycle-level microarchitectural simulator for DNN Inference Accelerator SIGMA Qin et al. (2020a) for this experiment. We use the docker image released by the authors of Muñoz-Matrínez et al. (2021). We use the standard configuration of SIGMA with 256 multiplier switches, 256 read ports in SDMemory and 256 write ports in SDMemory. The reduction networks is set to ASNETWORK and the memory controller is set to SIGMA_SPARSE_GEMM. We use SimulatedConv2d function in the PyTorch frontend version of STONNE. For a given convolutional layer, we run STONNE twice, once with 0% sparsity and once with 65% sparsity. We calculate the reduction in energy consumption by dividing the energy of the dense convolutional layer by the energy of the sparse convolutional layer. Since the weights' precision (or bit-width) is a parameter of SIGMA, the reduction in energy due to sparsity when compared to the dense model is not a function of the precision of the weights of the DNN.

## B    Visualizing Exploitation of Repetition and Sparsity

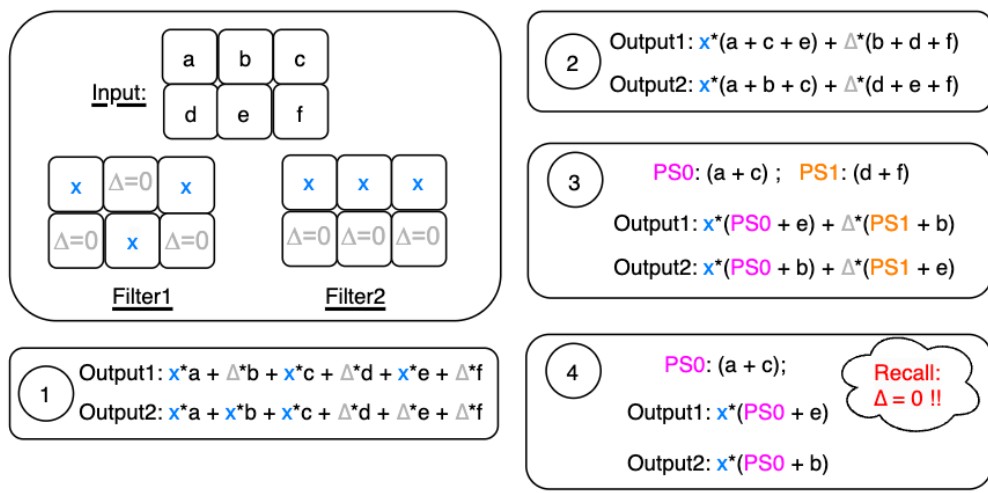

Figure 8: Visualization of repetition and sparsity during inference in modern systems: (1) Input and two filters that need to be multified (2) naive multiplication of weights and activations to create output 1 and output 2. (3) Re-ordering of weights and activations to simplify work within each filter (and exploit intra filter repetition phenomenon (Hegde et al., 2018)) (3) Using partial sums to reduce the work across filters (and exploit intra filter repetition phenomenon (Hegde et al., 2018)) (4) exploiting sparsity (by acknowledging repetition of zero weights as a special case of weight repetition (Sze et al., 2020; Hegde et al., 2018; Prabhakar et al., 2021).

## C    Experimental Setup Accuracy

**CIFAR10**   The data loader pipeline consists of simple augmentations - padding by 4 pixels on each size, random crop to $32 \times 32$, Random Horizontal Flip with probability 0.5, and normalization. We train from scratch for 350 epochs and use the Adam Optimizer Kingma & Ba (2014). We start with an initial learning

rate of 0.01 and reduce it by a factor of 10 at epochs 150, 200, and 320. For apples-to-apples comparison with binary and ternary, we do a sweep over batch sizes {16, 32, 64, 128, 256} and activation functions (ReLU, PReLU, TanH) and report the best top-1 validation accuracy. For ablations on (1) value assignment percentage and (2) comparison with binary networks with comparable effectual operations, we select the batch size the to be 32 and activation function to be PReLU. We compare the binary and signed-binary methods on their non-zero valued parameters of quantized layers as other aspects would be similar. When comparing against prior art works, we run these methods on our setup and report numbers. Further ablations on batch size and activation function provided in the appendix.

**ImageNet** We train ResNet-18 He et al. (2016) using SBWN on ImageNet Deng et al. (2009). We use standard practices used to train binary networks like (1) normalize the input using batch-norm Ioffe & Szegedy (2015) before convolution instead of after convolution Rastegari et al. (2016), (2) the first and the last layers are not quantized Zhu et al. (2016); Pouransari et al. (2020); Li et al. (2016); Bai et al. (2018). We use first order polynomial learning-rate annealing schedule with Adam optimizer Kingma & Ba (2014) along with PReLU He et al. (2015); Maas et al. (2013). We use FFCV dataloader Leclerc et al. (2022) with simple augmentations - Random Resize Crop to $224 \times 224$, Random Horizontal Flipping and Color Jitter with (brightness, contrast, saturation, hue) set as (0.4, 0.4, 0.4, 0). We decrease the learning rate from $2.0 \times e^{-4}$ to $2.0 \times e^{-8}$ while training for 320 epochs and do not use weight decay and batch size of 256 for training. We compare the binary and signed-binary methods on their non-zero valued parameters of quantized layers as other aspects would be similar. When comparing against prior art works, we report numbers as is from the literature due to high compute cost of running these experiments. Furthermore, for ResNet 34, we simply increase the model size.

**Implementation** Signed-Binary is a local binarization scheme, i.e., the quantization function takes full-precision latent weights of region of a convolutional layer as input and maps it to either $\{0, 1\}^{R \times S \times C_t}$ or $\{0, -1\}^{R \times S \times C_t}$. The values of the quantization function for these predetermined regions of a convolutional layer are determined randomly before training commences and remain unchanged. Different regions can have distinct quantization functions for a convolutional layer. This framework categorizes these regions into two buckets, optimizing efficiency by grouping them based on their quantization function values for more streamlined training. For example, if $C_t = C$, signed binarization becomes a per filter quantization scheme and each filter will have a different quantization function. We quantize the full-precision latent weights of a convolutional layer from $\mathbb{R}^{\{R \times S \times C \times K\}}$ to $\{0, 1\}^{R \times S \times C \times K \times P}$ and $\{0, -1\}^{R \times S \times C \times K \times (1-P)}$ where P is the percentage of filters whose quantization functions have the values $\{0,1\}$ such that $K \times P$ is an integer.

## D  Baselines

**Baselines for Comparison with Prior work**:  Figure 5 shows comparison against prior-art methods. The numbers are reported from the literature: DIR-Net Qin et al. (2023) LQ-Net Zhang et al. (2018), DSQ Gong et al. (2019), BC Courbariaux et al. (2015), ProxQuant Bai et al. (2018), IR-Net Qin et al. (2020b), DoReFA Zhou et al. (2016), SQ-BWN Dong et al. (2017), BWN Rastegari et al. (2016), BWNH Hu et al. (2018), ABC-Net Lin et al. (2017), BWHN Hu et al. (2018), LS Pouransari et al. (2020), Bi-Real Liu et al. (2018).

## E  Signed-Binary vs Binary wrt Effectual parameters

 We would like to compare binary with signed-binary when the DNN has the same number of non-zero parameters. Signed-Binary ResNet trained on CIFAR10 has slightly greater than 50% sparsity. If we reduce the total number of parameters of the binary ResNet by half, the resulting model would have comparable number of non-zero weights to signed-binary ResNet. This is done by reducing depth (see Table 7a) and by reducing width (see Table 7b). We train these models under identical conditions (setup details in the appendix). To clarify, row 1 & row 2 of Table 7 have the same number of total parameters while row 1 & row 3 have a comparable number of non-zero parameters. Thus, signed-binary leads to a higher accuracy than binary when both methods have comparable number of effectual operations.

| Quant | # Parameters | Depth | Acc |
|---|---|---|---|
| SB | 0.46M | 32 | 91.55% |
| B | 0.46M | 32 | 91.22% |
| B | 0.27M | 20 | 90.16% |

(a) **Reducing number of parameters by reducing depth**: We observe that accuracy of binary is 1.3% lower than signed-binary with comparable non-zero weights.

| Quant | # Parameters | Width | Acc |
|---|---|---|---|
| SB | 0.27M | $1\times$ | 90.05% |
| B | 0.27M | $1\times$ | 90.20% |
| B | 0.14M | $\lceil 0.7\times \rceil$ | 88.5% |

(b) **Reducing number of parameters by reducing width**: We observe that accuracy of binary is 1.7% lower than signed-binary with comparable non-zero weights.

Table 7: **Binary vs Signed-Binary with comparable non-zero weights**: We observe that Signed-Binary achieves higher accuracy when compared to binary with comparable effectual operations.

## F   Additional Ablations on CIFAR-10

| Batch Size | Accuracy (Top-1) |
|---|---|
| 16 | 89.44 |
| 32 | 90.05 |
| 64 | 89.62 |
| 128 | 89.59 |

(a) **Ablation on Batch Size**: The setup is identical across batch sizes and the non-linearity used is PReLU. We observe a decrease in accuracy when a high batch size of 256 is used.

| Non-Linearity | Accuracy (Top-1) |
|---|---|
| ReLU | 88.64 |
| PReLU | 90.05 |
| TanH | 88.75 |
| LReLU | 89.22 |

(b) **Ablation on Non-Linearity**: The setup is identical across non-linearity and the batch size used is 32. We observe that PReLU works best for our method.

Table 8: **Additional Ablations on CIFAR10**: Abations on batch size and non-linearity for SB.

We perform ablation on (1) batch sizes, (2) non-linearity on CIFAR10 (Krizhevsky & Hinton, 2009) dataset and report the numbers in Table 8a and  8b respectively. The setup is the same as mentioned above. We observe that for our method, there is a drop in accuracy with higher batch size, PReLU (Maas et al., 2013) works best and it is not sensitive to the choice of delta.

## G   Arithmetic Reduction Ablation

The fewer arithmetic operations required during inference for a given layer, the more efficient the quantization scheme is for both CPUs and GPUs (Hegde et al., 2018; Prabhakar et al., 2021; Fu et al., 2022). This can be measured using arithmetic reduction (Hegde et al., 2018; Prabhakar et al., 2021; Fu et al., 2022), which is defined as the ratio of arithmetic operations required using naive dense computation (unaware of repetition and sparsity) to the arithmetic operations taken during repetition-sparsity aware inference for a given DNN block. This metric indicates inference efficiency at the algorithmic level (Hegde et al., 2018; Prabhakar et al., 2021; Fu et al., 2022). Figure 9 compares the arithmetic reduction across different quantization schemes. The experiment follows the original test settings and methodologies described by the authors of (Prabhakar et al., 2021), using synthetic, uniformly distributed weights in the DNN block. The results show that signed-binary quantization is the most efficient, providing the highest arithmetic reduction across all conv layers.

We extend this experiment by varying the sparsity percentage from 0% to 100%, with equal percentages of positive and negative weights across all three quantization schemes. Figure 10 compares the arithmetic reduction (Y-axis) and the percentage of sparsity (X-axis) for a convolutional block of shape [3,3,512,512]. Since binary quantization uses +1, -1 and does not leverage sparsity, its performance is represented by a horizontal line. Ternary quantization initially behaves like binary when sparsity is negligible and performs

worse with moderate sparsity before improving under high sparsity conditions. This can be explained by the repetition-sparsity trade-off. Signed-binary quantization consistently outperforms ternary due to higher weight repetition with equal sparsity and outperforms binary due to the presence of sparsity. The highly efficient behavior of signed-binary under both very low and very high sparsity can be explained: with approaching 0% sparsity, signed binary gets converted into monolithic filters containing one unique weight per filter, and with high sparsity, most operations become ineffectual, resulting in significant savings.

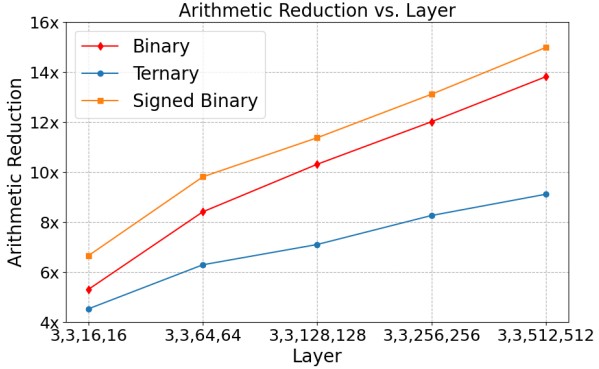

Figure 9: Arithmetic Reduction (higher is better) for Binary, Ternary, and Signed-Binary across different DNN blocks.

Figure 10: Arithmetic Reduction (higher is better) for Binary, Ternary, and Signed-Binary across different degrees of sparsity.

## H  Latent Full-Precision Weights & Standardization

Binary quantization has shown to improve accuracy from standardization of latent full-precision weights. However, this trend is not observed in signed-binary networks.

| Standardization Strategy | Accuracy (%) |
|---|---|
| Local Signed-Binary Regions | 59.1 |
| Global Signed-Binary Block | 61.2 |
| No Standardization | 61.4 |

Table 9: Comparison of Standardization Strategies on Accuracy

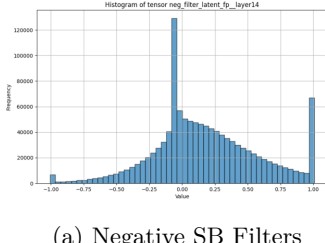
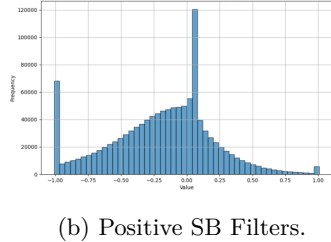
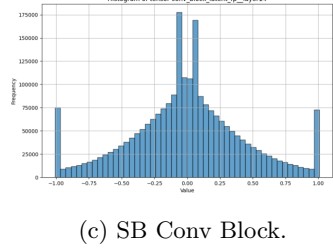

(a) Negative SB Filters        (b) Positive SB Filters.        (c) SB Conv Block.

Figure 11: Distribution of Latent Full-Precision Weights: While the local signed-binary weights are neither zero mean, nor Laplacian, the entire conv block is zero mean Laplacian distribution. The peaks in the image are because of clipping at $\pm 1$ along with thresholding at $\pm \Delta$.

## I  Additional ImageNet Ablations

We provide additional ImageNet ablations by training ResNet-18 architecture.

| %{0,1} filters | %{0,-1} filters | Acc |
|---|---|---|
| 1 | 0 | 55.23 |
| 0.25 | 0.75 | 61.94 |
| 0.5 | 0.5 | 62.29 |

Table 10: Accuracy with Filters

| EDE† | Acc |
|---|---|
| Disabled | 62.73 |
| Enabled | 63.17 |

Table 11: EDE† Accuracy

| $Delta \ \Delta$ | Acc |
|---|---|
| $0.01 \times \max |W|$ | 64.1 |
| $0.05 \times \max |W|$ | 64.3 |

Table 12: Delta Accuracy

## J  Datasets

| Dataset | License | Source |
|---|---|---|
| ImageNet | Non-Commercial | ILSVRC2012 |
| CIFAR10 | N/A | CIFAR |

Table 13: **Dataset with Licenses**: License and source of the datasets used.

Licenses of ImageNet (Deng et al., 2009) and CIFAR10 (Krizhevsky & Hinton, 2009) datasets used in this paper are listed in Table 1. Every accuracy reported in this paper is on validation set of the dataset. ImageNet and CIFAR10 are standard publicly used datasets. Since they do not own their images, therefore they do not have a release license. Actual images may have their own copyrights but ImageNet provides stipulations for using their dataset (for non-commercial use). We do not recommend using the resulting signed-binary models trained on these datasets for any commercial use.

