# OpenReview forum: "PLUM: Improving Inference Efficiency By Leveraging Repetition-Sparsity Trade-Off"
_TMLR — Accepted by TMLR_

### Review · Reviewer_3Yto · 2024-07-10

**Summary Of Contributions:**

In this paper, the authors introduce the concept of repetition-sparsity trade-off, based on which the proposed method PLUM is designed, which integrates DNN inference systems and quantization to improve inference efficiency. Experiments show that PLUM retains model accuracy with fewer effectual parameters due to its signed-binary quantization scheme. Demonstrations on Intel CPUs show that PLUM improves inference speed and energy efficiency.

**Audience:**

Yes

**Claims And Evidence:**

No

**Requested Changes:**

The requested changes in the following comments are critical for acceptance.
1. In the introduction, the novelty and contributions are not clearly stated. The concept of repetition-sparsity trade-off seems straightforward, but how is PLUM designed based on this concept? The contributions and novelty should be made more clear. It is now lacking in the introduction.
2. Following the above comment, there is a gap between the repetition-sparsity trade-off concept and the proposed scheme. In what sense is PLUM an optimal solution to solve this trade-off? A mathematically rigorous analysis is required, otherwise, the method design looks ad-hoc.
3. A concept of repetition-sparsity trade-off is essential in this paper. Is there any similar concept mentioned in previous works, or is it the first time this concept is introduced? A comprehensive literature review is required to show how this concept of repetition-sparsity trade-off differs from the previous works.
4. The references in Section 2 are mostly published before 2020, more recent references (published in 2023 and 2024) should be included, especially when the reference is called "recent work".
5. the ablation for PLUM comes out of a sudden in Section 3.2.2. it should be included in the experimental part to avoid distraction. Moreover, the explanation for this ablation is missing.
6. In Figure 5, what are the methods that PLUM is compared to? Please carefully cite and explain the methods used for comparison.
6. Appendix B is empty? At least some explanation or description is needed. Fig. 8 is not even referred to.

**Strengths And Weaknesses:**

Strengths:
1. The proposed PLUM framework is a relatively simple yet effective approach that considers repetition-sparsity trade-off in the DNN inference process. Experiments validate its accuracy retaining with less effectual parameters.
2. The paper provides an analysis of the experiments of PLUM, providing detailed charts and figures.

Weaknesses:
1. The contribution of the paper is not fully analyzed. There is a gap between the concept of repetition-sparsity trade-off and the proposed scheme.
2. Experiments are not convincing enough, where baseline methods and models for evaluation are limited.
3. The paper is not clearly presented and organized.
4. Figures and tables are not clearly explained.

---

> ### Author Response · Authors · 2024-07-25
> **Author Response (1/2)**
>
> We thank the reviewer for the detailed evaluation and comments. We thank the reviewer for giving his perspective. The responses are not in line with the requested changes. We have mapped responses with the requested changes using “RC-X”, where X is the requested change from the reviewer’s list. Responses are in line with the updated PDF.
>
> **[RC1] Novelty and contributions are not apparent and are absent from the introduction.**
>
> We thank the reviewer for the comment. We refer the reviewer to **Section 1, Introduction, on top of Page 3 for a list of contributions** of this work (we are slightly confused about whether it is not clear to the reviewer).
>
> We would be thankful if the reviewer gave us a chance to highlight our contributions again:
>
> 1. The key insight of this work is the repetition-sparsity trade-off, which explains the inefficiency of prior work.
> 2. The proposed PLUM framework aims to demonstrate and leverage the existence of repetition-sparsity trade-off, resulting in improved inference efficiency while retaining model accuracy.
> 3. PLUM’s signed-binary quantization retains model accuracy (90.7% and 66.2% on CIFAR and ImageNet) while requiring significantly fewer effectual parameters.  We offer detailed insights by visualizing the distribution of latent full-precision and quantized weights when using PLUM.
> 4. We demonstrate inference efficiency with respect to latency, energy consumption, and model density.
>
> **[RC6,4,5] Provide more recent references in Section 2. Are baselines listed in Section 2?**
>
> 1. We thank the reviewer for the comment. **Supplementary D** lists the baselines for Figure 5. The numbers in Figure 5 are reported from the literature.
> 2. Since our work lies at the intersection of two different research areas of computer architecture and machine learning, we originally intended **Section 2**, titled Background, to give fundamental ideas to a diverse audience. We have updated Section 2 to add references from **Supplementary D**, which are more recent as requested.
> 3. Our writing style is inspired by [8].
>
> **[RC1,3] Is repetition-sparsity trade-off an original contribution of this work?**
>
> Yes.
>
> **[RC3] Provide a literature review to show how the concept of repetition-sparsity trade-off differs from previous works.**
>
> We thank the reviewer for the comment. Updated **Section 2** talks about how prior work is ignorant of the repetition-sparsity trade-off (which is an original contribution of this work). We would be thankful if the reviewer gave us a chance to highlight why repetition-sparsity trade-off does not exist in literature again:
>
> 1. Prior quantization works [1,2,3,4] are focused almost exclusively on model accuracy and are ignorant of weight repetition and weight sparsity with respect to efficiency.
> 2. Prior DNN inference works [5,6,7,9] are built for trained quantized DNNs using existing quantization schemes [1,2,3, 4]. Thus they treat repetition and sparsity to be rigid leading to suboptimal inference performance..
> 3. We are the first to co-design while considering model accuracy, weight repetition, and sparsity, i.e., propose repetition-sparsity trade-off as key insight and design PLUM.
>
> **[RC1] How does PLUM leverage the key insight of this work, i.e., repetition-sparsity trade-off?**
>
> We thank the reviewer for the comment. We would be thankful if the reviewer gave us a chance to explain this again step-by-step by using Section 3 and Figure 4 in the paper:
> 1. **Section 3.1** defines repetition-sparsity trade-off: We can use any of the four quantization functions ({1,-1}, {1,0,-1}, {1,0}, {-1,0}) for a given latent full-precision weight in a DNN block. This variety of assignments to individual weights allows for the creation of unique configurations within the block, each representing a distinct point along the repetition-sparsity spectrum.
> 2. **Section 3.1**: The PLUM framework aims to identify a more efficient point on the repetition-sparsity spectrum. The objective for PLUM is to improve efficiency while retaining accuracy.
> 3. **Section 3.2.1 and Figure 4**: During DNN inference, PLUM is designed such that a single processing step during inference sees one signed binary quantization function. Thus reaping the benefits of binary while introducing sparsity, thus leveraging repetition sparsity trade-off.
> 4. **Section 3.2.1**: This requires the PLUM quantization scheme to be locally binary while globally ternary for a DNN block.
> 5. **Section 3.2.2**: PLUM’s quantization signed-binary is methodologically designed to support the above-mentioned constraints using ablations on the CIFAR10 dataset.
>
> **[RC2] Is PLUM an optimal solution to solve this trade-off?**
>
> We thank the reviewer for the comment. PLUM aims to leverage the repetition-sparsity tradeoff (and thus prove its existence) to get inference efficiency. We do not claim that it is a mathematically optimal solution to solve this trade-off. The proposed framework is empirically designed using ablations.

---

> > ### Author Response · Authors · 2024-07-25
> > **Author Response (2/2)**
> >
> > **[RC2] In what sense is PLUM a more efficient solution to solve this trade-off?**
> >
> > We encourage the reviewer to read **Common Response 2** before reading this forward. Prior work [5,6,7] shows that the fewer the arithmetic operations required for repetition-sparsity aware inference, the more efficient the quantization schema is. We have extended Supplementary G to include the plots shown below, along with extended Section 5.1. We can conclude that signed-binary requires fewer operations (and thus more efficient) in all three settings: (1) across different conv layers, (2) across different percentages of sparsity, and (3) for real DNNs trained on ImageNet.
> >
> > **Please refer to Figure 9 in Supplementary G.** Table is corresponding to the same:
> >
> > *Arithmetic Reduction vs Layer*:
> >
> > | Layer Size      | Binary | Ternary | Signed-Binary |
> > |--------|--------|---------|------|
> > | [3,3,16,16]     | 5.3x   | 4.52x   | **6.65x**  |
> > | [3,3,64,64]     | 8.4x   | 6.28x   | **9.8x**    |
> > | [3,3,128,128]   | 10.3x  | 7.09x   | **11.36x**    |
> > | [3,3,256,256]   | 12x    | 8.25x   | **13.1x**   |
> > | [3,3,512,512]   | 13.8x  | 9.1x    | **14.97x**   |
> >
> >
> > We extend this experiment by varying the sparsity percentage from 0% to 100%, with equal percentages of positive and negative weights across all three quantization schemes. **Figure 10** compares the arithmetic reduction (Y-axis) and the percentage of sparsity (X-axis) for a convolutional block of shape [3,3,512,512].
> >
> > 1. Since binary quantization uses {+1, -1} and does not leverage sparsity, its performance is represented by a horizontal line.
> > 2. Ternary quantization initially behaves like binary when sparsity is negligible and performs worse with moderate sparsity before improving under high sparsity conditions.  This can be explained by the repetition-sparsity trade-off.
> > 3. Signed-binary quantization consistently outperforms ternary due to higher weight repetition with equal sparsity and outperforms binary due to the presence of sparsity. The highly efficient behavior of signed-binary under both very low and very high sparsity can be explained: with approaching 0% sparsity, signed binary gets converted into filters containing either only {+1} or {-1} weights, and with high sparsity, most operations become ineffectual, resulting in significant savings.
> >
> > **Please refer to Figure 10 in Supplementary G.** Table is corresponding to the same:
> >
> > *Arithmetic Reduction vs Sparsity*:
> >
> > | Percentage of Zero Weights (%) | Ternary | Binary | Signed Binary |
> > |----------|---------|--------|---------------|
> > | 1   | 12.64x | 13.99x | **43.72x** |
> > | 5   | 10.23x | 13.99x | **32.44x** |
> > | 10  | 9.46x  | 13.99x | **24.74x** |
> > | 20  | 9.03x  | 13.99x | **16.98x** |
> > | 30  | 9.08x  | 13.99x | **15.03x** |
> > | 40  | 9.13x  | 13.99x | **14.86x** |
> > | 50  | 9.25x  | 13.99x | **14.96x** |
> > | 60  | 9.75x  | 13.99x | **14.97x** |
> > | 70  | 11.28x | 13.99x | **15.80x** |
> > | 80  | 15.17x | 13.99x | **18.95x** |
> > | 90  | 23.97x | 13.99x | **30.49x** |
> >
> > **[RC7] Appendix B.**
> >
> > Noted. We fixed the formatting to move Figure 8 to Appendix B.
> >
> > **References**
> >
> > [1] Qin, Haotong, et al. "Distribution-sensitive information retention for accurate binary neural network." International Journal of Computer Vision 131.1 (2023): 26-47.
> >
> > [2] Qin, Haotong, et al. "Forward and backward information retention for accurate binary neural networks." Proceedings of the IEEE/CVF conference on computer vision and pattern recognition. 2020.
> >
> > [3] Bai, Yu, Yu-Xiang Wang, and Edo Liberty. "ProxQuant: Quantized Neural Networks via Proximal Operators." International Conference on Learning Representations.
> >
> > [4] Zhang, Dongqing, et al. "Lq-nets: Learned quantization for highly accurate and compact deep neural networks." Proceedings of the European conference on computer vision (ECCV). 2018.
> >
> > [5] Hegde, Kartik, et al. "UCNN: Exploiting computational reuse in deep neural networks via weight repetition." 2018 ACM/IEEE 45th Annual International Symposium on Computer Architecture (ISCA). IEEE, 2018.
> >
> > [6] Prabhakar, Rohan Baskar, et al. "Summerge: An efficient algorithm and implementation for weight repetition-aware dnn inference." Proceedings of the ACM International Conference on Supercomputing. 2021.
> >
> > [7] Fu, Cheng, et al. "Q-gym: An equality saturation framework for dnn inference exploiting weight repetition." Proceedings of the International Conference on Parallel Architectures and Compilation Techniques. 2022.
> >
> > [8] Liu, Zhuang, Hanzi Mao, Chao-Yuan Wu, Christoph Feichtenhofer, Trevor Darrell, and Saining Xie. "A convnet for the 2020s." In Proceedings of the IEEE/CVF conference on computer vision and pattern recognition, pp. 11976-11986. 2022.
> >
> > [9] Cowan, Meghan, Thierry Moreau, Tianqi Chen, James Bornholt, and Luis Ceze. "Automatic generation of high-performance quantized machine learning kernels." In Proceedings of the 18th ACM/IEEE International Symposium on Code Generation and Optimization, pp. 305-316. 2020.

---

### Review · Reviewer_WV4c · 2024-07-11

**Summary Of Contributions:**

This paper highlights that binary quantization is an extreme form of repetition, often overlooking weight sparsity. In contrast, ternary quantization introduces weight sparsity but results in reduced efficiency. To address this issue, the authors propose a trade-off quantization scheme. Additionally, they co-designed the quantization algorithm and the inference kernel. Two strengths are as follows:
1. The idea is intriguing and logical; such a trade-off scheme can push the boundaries of quantization.
2. The proposed quantization successfully introduces sparsity into binary quantization while maintaining accuracy and speeding up inference.

**Audience:**

Yes

**Claims And Evidence:**

Yes

**Requested Changes:**

see the weaknesses for details

**Strengths And Weaknesses:**

1. Minor: The right part of Figure 3 contains two sparsity items.
2. As indicated in Eq(2), each weight requires a sign-factor and a bitmap. Therefore, is each weight represented by 2 bits? If so, the advantage of PLUM partially comes from more efficient utilization of bits compared to ternary quantization. However, the model size for PLUM is also double that of binary quantization.
3. GPUs are optimized for parallel computation. A potential drawback of this scheme is that unstructured sparsity may hinder inference speedup. in GPU

---

> ### Author Response · Authors · 2024-07-25
>
> We thank the reviewer for the detailed evaluation and comments. We thank the reviewer for giving his perspective. The following clarifications are inline with the updated version of the paper:
>
> **Figure 3**
>
> We thank the reviewer for the comment. We have fixed the figure.
>
> **Bit representation for the signed binary.**
>
> We thank the reviewer for the comment. We have added a discussion in **Section 6**. Let the convolutional layer have an R × S kernel size with C input channels and K filters.
>
> 1. R * S * C * K bits for binary
> 2. R * S * C * K * 2 bits for ternary
> 3. R * S * C * K + K bits for signed-binary: This is because we need one additional bit per filter to signify to which signed-binary quantization function it belongs to. We can perform two’s complement when a filter corresponding to {-1,0} quantization function is used.
>
> We encourage the reviewer to see **Figure 4** to visualize this better. In modern systems [6,7], only once all activations are added is the partial sum multiplied by the unique weights. Thus, we don’t need R * S * C * K * 2 bits for signed binary but rather R * S * C * K + K bits.
>
> **Suitability of parallel computation.**
>
> We thank the reviewer for the comment. Please see **Common Response 2** for a detailed response. Essentially, prior works [6,7] have shown that modern systems are scalable in both CPUs and GPUs. In line with previous work, we use arithmetic operations required for DNN inference to demonstrate efficiency at the algorithmic level.
>
> **Appendix B**
>
>  We thank the reviewer for the comment. We have fixed the formatting.
>
> [6] Prabhakar, Rohan Baskar, et al. "Summerge: An efficient algorithm and implementation for weight repetition-aware dnn inference." Proceedings of the ACM International Conference on Supercomputing. 2021.
>
> [7] Fu, Cheng, et al. "Q-gym: An equality saturation framework for dnn inference exploiting weight repetition." Proceedings of the International Conference on Parallel Architectures and Compilation Techniques. 2022.

---

### Review · Reviewer_JW4J · 2024-07-12

**Summary Of Contributions:**

1. The authors introduce the concept of repetition-sparsity trade-off to explain the inefficiencies in inference performance of binary and ternary quantization methods.
2. The authors propose PLUM, a quantization-system co-design framework that leverages the repetition-sparsity trade-off to improve inference efficiency while maintaining model accuracy.
3. Through extensive experiments, the authors demonstrate the effectiveness of the PLUM framework:
a) PLUM's signed-binary quantization scheme achieves accuracy comparable to existing quantization methods on CIFAR10 and ImageNet datasets while using fewer effectual parameters.
b) Visualization analysis reveals the characteristics of the quantized weight distribution in PLUM and its relationship with the latent full-precision weights.
c) Inference experiments on Intel CPUs, measuring latency and energy consumption, validate the existence of the repetition-sparsity trade-off and showcase PLUM's advantages in accelerating inference speed and reducing energy consumption.
These experimental results indicate that the PLUM framework can significantly improve model inference performance while maintaining accuracy.

**Audience:**

Yes

**Claims And Evidence:**

Yes

**Requested Changes:**

Critical changes:

1. Conduct more comprehensive ablation studies on a larger scale to strengthen the empirical evidence supporting the effectiveness of the PLUM framework. This will help address the concern about the limited scale of the current ablation experiments and increase the persuasiveness of the results.
2. Provide a more detailed discussion on the limitations of the PLUM framework, including any potential trade-offs between model size, accuracy, and inference efficiency. This will help readers better understand the scope and applicability of the proposed method.
3. Include a detailed analysis of the training overhead of PLUM compared to other quantization methods, and discuss the potential impact on the practical adoption of the framework. This information is crucial for practitioners considering the use of PLUM in real-world scenarios.

Suggested improvements:

4. Evaluate the performance of the PLUM framework on other tasks (e.g., object detection, segmentation) and with different network architectures to demonstrate its generalizability. This will strengthen the paper's contribution and broaden its impact.
5. Provide more insights into the selection of hyperparameters for the PLUM framework, such as the choice of quantization function value sets and their impact on performance. This will help readers better understand how to apply the proposed method effectively.
6. Discuss the potential challenges in deploying PLUM on other hardware platforms (e.g., GPUs, edge devices) and suggest possible optimizations. This will provide valuable information for practitioners looking to implement the framework on different hardware setups.

**Strengths And Weaknesses:**

Strengths:
1. The concept of repetition-sparsity trade-off provides a novel and insightful perspective for analyzing existing quantization methods, helping to understand these methods' limitations better.
2. The PLUM framework demonstrates the importance of co-designing quantization methods with inference systems, offering a new approach for developing efficient inference solutions.
3. Through comprehensive experiments, the authors thoroughly demonstrate the effectiveness of the PLUM framework, including accuracy comparisons on standard benchmark datasets, visualization analysis of weight distributions, and inference performance evaluations on real hardware platforms.
4. The visualization analysis in the paper provides interesting insights into the learned features of the PLUM quantization scheme, helping readers better understand how the method works.
5. The inference experiments on Intel CPUs not only prove the effectiveness of the PLUM framework but also provide empirical evidence for the repetition-sparsity trade-off, strengthening the core argument of the paper.

Weaknesses:
1. The ablation studies for PLUM are conducted on a limited scale, which weakens the persuasiveness of the experiments. More comprehensive ablation studies on a larger scale would strengthen the empirical evidence supporting the effectiveness of the proposed method.
2. While the authors demonstrate the effectiveness of PLUM on image classification tasks using ResNet architectures, it would be beneficial to see the framework's performance on other tasks (e.g., object detection, segmentation) and with different network architectures to assess its generalizability.
3. The paper could benefit from a more detailed discussion of the limitations of the PLUM framework, such as any potential trade-offs between model size, accuracy, and inference efficiency.
4. The authors could provide more insights into the selection of hyperparameters for the PLUM framework, such as the choice of quantization function value sets and the impact of different choices on performance.
5. The paper would benefit from a discussion on the potential challenges in deploying PLUM on other hardware platforms (e.g., GPUs, edge devices) and any specific optimizations that might be required.
6. While the authors mention that PLUM requires training from scratch, which can be time-consuming and computationally expensive, they do not provide a detailed analysis of the training overhead compared to other quantization methods or the potential impact on the practical adoption of the framework.

---

> ### Author Response · Authors · 2024-07-25
>
> We thank the reviewer for the detailed evaluation and comments. We thank the reviewer for giving his perspective. The following clarifications are inline with the updated version of the paper:
>
> **More comprehensive ablations**
>
> We thank the reviewer for the comment. Please see **common response 1**. We have added additional experiments in **Supp I and J**.
>
> **Provide a more detailed discussion on the limitations of the PLUM framework.**
>
> We thank the reviewer for the detailed comments on our work. Following your request, we have added a discussion in **Section 6** on the limitations of the PLUM framework. Since PLUM is the first work that leverages the repetition-sparsity trade-off, we observe that efficiency metrics (latency, arithmetic operations, energy consumption, model density) improve while retraining accuracy when using PLUM. We summarize the  limitations here:
> 1. PLUM requires the use of two quantization functions, i.e., with value sets of {1,0} and {-1,0}. Only using a single quantization function (just {1,0}) could lead to improving inference efficiency. But we observe simply using {1,0} leads to an accuracy drop.
> 2. As claimed in S3.2.1, The tile size of the modern inference system should be set such that a single processing step should see only one signed binary quantization function.
>
> **Discuss constraints on the practical adoption of the framework.**
>
> Thank you for the comment. We have added a discussion in **Section 6**. There is no significant overhead in terms of training using PLUM as compared to conventional binary. It took one day of training on 8 16 GB GPUs. All hyperparameters are given in Supp C, and we will release code and checkpoints. We understand that commenting on training time is necessary for practical adoption. Further, we would like to highlight that the goal of this work is to show that PLUM can be leveraged for more efficient inference than conventional binary with similar accuracy. As we are measuring runtime efficiency, the training cost is not a measure of direct comparison for our work.
>
> **Evaluate performance on other datasets and architectures**
>
> We thank the reviewer for the comment. Please see **common response 1**. We have provided new experiments in **Supp J** to strengthen the paper’s core contributions.
>
> **Provide more hyperparameter insights.**
>
> We thank the reviewer for the comment. Please see **common response 1**. All hyperparameters in **Supplementary A and C**. We hope additional experiments provide readers with more insights, as the reviewer intended.
>
> **Hardware generalizability.**
>
> We thank the reviewer for the comment. Please refer to **Common Response 2** for a detailed response.

---

### Review · Reviewer_bHEV · 2024-07-12

**Summary Of Contributions:**

This paper introduces PLUM, a unified co-design framework that integrates DNN inference systems and quantization to leverage the repetition-sparsity trade-off to improve inference efficiency. Experiment results demonstrate more accuracy quantization over binary quantization with the same number of non-zero weights and 26% speedup on real hardware and doubled energy efficiency.

**Audience:**

Yes

**Broader Impact Concerns:**

N.A

**Claims And Evidence:**

Yes

**Requested Changes:**

See weakness

**Strengths And Weaknesses:**

Strengths:
1. The approach presents a simple-yet-effective framework, instead of a overly complicated design.
2. I think the experiments are complete and comprehensive.

Weakness:
1. The models used in experiments seem to be small scale. I wonder would it be possible to evaluate on larger scale of the models?
2. Figure presentation: Figure 3 seems to be a bit blurry; Figure 5 has overlapped legends.
3. Need to check for Typos. Appendix B is empty to me.

---

> ### Author Response · Authors · 2024-07-25
>
> We thank the reviewer for the detailed evaluation and comments. We thank the reviewer for giving his perspective. The following clarifications are inline with the updated version of the paper:
>
> **Models used in experiments are small-scale.**
>
> We thank the reviewer for the comment. Please refer to **common response 1**. Based on your comments, we have added additional experiments in **Supp I and J**.
>
> **Figures 3 and 5.**
>
> Thank you for pointing it out. We have updated the figures.
>
> **Appendix B is empty.**
>
> Noted. We have updated the supplementary and moved Fig 8 to Supp B.

---

### Review · Reviewer_KJZp · 2024-07-14

**Summary Of Contributions:**

This paper introduces PLUM, a framework that integrates DNN inference systems with quantization to leverage a repetition-sparsity trade-off for improved efficiency. PLUM’s quantization method is more accurate than binary quantization, and signed binarization effectively reduces non-zero parameters within DNNs. The framework achieves a 26% speedup, doubles energy efficiency, and reduces density by 2.8× compared to binary methods, while maintaining a 66.2% top-1 accuracy on ImageNet for ResNets. PLUM offers a viable solution for deploying efficient models in resource-limited environments.

**Audience:**

Yes

**Broader Impact Concerns:**

No.

**Claims And Evidence:**

Yes

**Requested Changes:**

See the weaknesses above.

**Strengths And Weaknesses:**

Strengths:
1. This paper is well written and easy to follow.
2. The proposed new binary representation is interesting and seems very promising.

Weaknesses:
1. Most of the experiments are conducted on CIFAR-10 which greatly undermined the credibility of the method.
2. Speed up experiments seems only conducted on CPU, No GPU results.
3. How to combine with activation binarization. Can PLUM be adopted in FPGA as vanilla BNN?
4. The training time and cost is not clear too.

---

> ### Author Response · Authors · 2024-07-25
>
> We thank the reviewer for the detailed evaluation and comments. We thank the reviewer for giving his perspective. The following clarifications are inline with the updated version of the paper:
>
> **Most experiments are conducted on CIFAR-10.**
>
> We thank the reviewer for the comment. Please see **common response 1**. We have added additional experiments in **Supp I and J**.
>
> **Hardware generalizability.**
>
> We thank the reviewer for the comment that has prompted us to run additional experiments showcasing the effectiveness of PLUM. Please see **common response 2**. We have extended S5.1 to demonstrate arithmetic reduction numbers along with extending **supplementary G**.
>
> Essentially, prior work has demonstrated the scalability of SOTA repetition-sparsity aware inference algorithms on CPUs [6], GPUs [7], and ASICs [5]. In line with prior work, we use arithmetic operations required for DNN inference to demonstrate efficiency at the algorithmic level.
>
> **Can PLUM be adopted in FPGA as vanilla BNN?**
>
> Great suggestion. The goal of this work is to propose and showcase the importance of repetition-sparsity tradeoff and highlight PLUM as a solution for the same. Implementing it in a custom FPGA is out of the scope of this work. However, we will release PLUM’s code and checkpoints. We implore the community to take it up as future work.
>
> **Training time and cost?**
>
> Thank you for the comment. We have added a discussion in **Section 6**. There is no significant overhead in terms of training using PLUM as compared to conventional binary. It took one day of training on 8 16 GB GPUs. All hyperparameters are given in **Supp A and C**. Further, we would like to highlight that the goal of this work is to show that PLUM can be leveraged for more efficient inference than conventional binary with similar accuracy. As we are measuring runtime efficiency, the training cost is not a measure of direct comparison for our work.
>
> [5] Hegde, Kartik, et al. "UCNN: Exploiting computational reuse in deep neural networks via weight repetition." 2018 ACM/IEEE 45th Annual International Symposium on Computer Architecture (ISCA). IEEE, 2018.
>
> [6] Prabhakar, Rohan Baskar, et al. "Summerge: An efficient algorithm and implementation for weight repetition-aware dnn inference." Proceedings of the ACM International Conference on Supercomputing. 2021.
>
> [7] Fu, Cheng, et al. "Q-gym: An equality saturation framework for dnn inference exploiting weight repetition." Proceedings of the International Conference on Parallel Architectures and Compilation Techniques. 2022.

---

### Author Response · Authors · 2024-07-25
**Common Response (1/2)**

We thank the reviewers for their insightful comments and valuable feedback. We have uploaded the updated version of the paper. The following clarifications are in line with the updated version of the paper:

**C1 Accuracy-related ablation experiments are focussed on CIFAR10. Give more experiments.**

We thank the reviewers for the question. We follow the convention set up by prior work [1,2,3,4] of performing ablations using CIFAR10 and then scaling to ImageNet (which is computationally expensive). However, we have added supplementary I for additional ablations on ImageNet and supplementary J for additional dataset ablations.

Please see **Supplementary I**. We summarize the additional ImageNet ablations results here for the reviewers:
| %{0, 1} filters |  %{0, −1} filters | Accuracy |
|----------|----------|----------|
| 100        | 0       | 55.23    |
| 25        | 75       | 61.94    |
| 50     | 50     | 62.29    |

| EDE       | Accuracy |
|-----------|----------|
| Disabled  | 62.73    |
| Enabled   | 63.17    |

| Delta (∆)      | Accuracy |
|----------------|----------|
| 0.01 × max(abs(W)) | 64.1     |
| 0.05 × max (abs(W)) | 64.3     |

Please see **Supplementary H**. Further ImageNet ablation for weight standardization:

| Standardization Strategy        | Accuracy |
|---------------------------------|----------|
| Local Signed-Binary Regions     | 59.1     |
| Global Signed-Binary Block      | 61.2     |
| No Standardization              | 61.4     |

Please see **Supplementary J**. We summarize the signed binary ablations results here for the reviewers:

| Model     | Dataset      | Accuracy Signed Binary | Accuracy Full Precision |
|-----------|--------------|------------------------|-------------------------|
| ResNet20  | CIFAR10      | 90.05                  | 92.10                   |
| ResNet32  | CIFAR10      | 91.55                  | 92.90                   |
| ResNet44  | CIFAR10      | 91.98                  | 93.30                   |
| ResNet56  | CIFAR10      | 92.52                  | 93.63                   |
| ResNet110 | CIFAR10      | 92.68                  | 93.83                   |
| VGG       | CIFAR10      | 92.90                  | 93.80                   |
| AlexNet   | SVHN         | 97.20                  | 97.70                   |
| ResNet18  | CIFAR100     | 75.83                  | 77.80                   |
| ResNet18  | TinyImageNet | 56.90                  | 59.72                   |
| ResNet18  | ImageNet     | 66.20                  | 69.50                   |
| ResNet34  | ImageNet     | 70.50                  | 73.10                   |

[1] Qin, Haotong, et al. "Distribution-sensitive information retention for accurate binary neural network." International Journal of Computer Vision 131.1 (2023): 26-47.

[2] Qin, Haotong, et al. "Forward and backward information retention for accurate binary neural networks." Proceedings of the IEEE/CVF conference on computer vision and pattern recognition. 2020.

[3] Bai, Yu, Yu-Xiang Wang, and Edo Liberty. "ProxQuant: Quantized Neural Networks via Proximal Operators." International Conference on Learning Representations.

[4] Zhang, Dongqing, et al. "Lq-nets: Learned quantization for highly accurate and compact deep neural networks." Proceedings of the European conference on computer vision (ECCV). 2018.

---

> ### Author Response · Authors · 2024-07-25
> **Common Response (2/2)**
>
> **C2 Hardware generalizability. Comment.**
>
> We thank the reviewers for the question. We have expanded **Supp G and S5.1** to explain this better in the updated manuscript. We would be thankful if the reviewers give us a chance to explain this in detail:
>
> Please see **Supp G**. To understand hardware generalizability more broadly, we must understand how prior art [5,6,7] disentangles repetition-sparsity-aware DNN inference algorithms from their implementations. The fewer arithmetic operations required during inference for a given layer, the more efficient the quantization scheme is for both CPUs and GPUs [5,6,7]. This can be measured using arithmetic reduction [5,6,7], which is defined as the ratio of arithmetic operations required using naive dense computation (unaware of repetition and sparsity) to the arithmetic operations taken during repetition-sparsity aware inference for a given DNN block. This metric indicates inference efficiency at the algorithmic level [5,6,7]. **Figure 9** compares the arithmetic reduction across different quantization schemes. The experiment follows the original test settings and methodologies described by the authors of [6], using synthetic, uniformly distributed weights in the DNN block. The results show that signed-binary quantization is the most efficient, providing the highest arithmetic reduction across all convolutional layers.
>
> **Please refer to Figure 9 in Supplementary G.** Table is corresponding to the same:
>
> *Arithmetic Reduction vs Layer*:
>
> | Layer Size      | Binary | Ternary | Signed-Binary |
> |--------|--------|---------|------|
> | [3,3,16,16]     | 5.3x   | 4.52x   | **6.65x**  |
> | [3,3,64,64]     | 8.4x   | 6.28x   | **9.8x**    |
> | [3,3,128,128]   | 10.3x  | 7.09x   | **11.36x**    |
> | [3,3,256,256]   | 12x    | 8.25x   | **13.1x**   |
> | [3,3,512,512]   | 13.8x  | 9.1x    | **14.97x**   |
>
>
> We extend this experiment by varying the sparsity percentage from 0% to 100%, with equal percentages of positive and negative weights across all three quantization schemes. **Figure 10** compares the arithmetic reduction (Y-axis) and the percentage of sparsity (X-axis) for a convolutional block of shape [3,3,512,512].
>
> 1. Since binary quantization uses {+1, -1} and does not leverage sparsity, its performance is represented by a horizontal line.
> 2. Ternary quantization initially behaves like binary when sparsity is negligible and performs worse with moderate sparsity before improving under high sparsity conditions.  This can be explained by the repetition-sparsity trade-off.
> 3. Signed-binary quantization consistently outperforms ternary due to higher weight repetition with equal sparsity and outperforms binary due to the presence of sparsity. The highly efficient behavior of signed-binary under both very low and very high sparsity can be explained: with approaching 0% sparsity, signed binary gets converted into filters containing either only {+1} or {-1} weights, and with high sparsity, most operations become ineffectual, resulting in significant savings.
>
> **Please refer to Figure 10 in Supplementary G.** Table is corresponding to the same:
>
> *Arithmetic Reduction vs Sparsity*:
>
> | Percentage of Zero Weights (%) | Ternary | Binary | Signed Binary |
> |----------|---------|--------|---------------|
> | 1   | 12.64x | 13.99x | **43.72x** |
> | 5   | 10.23x | 13.99x | **32.44x** |
> | 10  | 9.46x  | 13.99x | **24.74x** |
> | 20  | 9.03x  | 13.99x | **16.98x** |
> | 30  | 9.08x  | 13.99x | **15.03x** |
> | 40  | 9.13x  | 13.99x | **14.86x** |
> | 50  | 9.25x  | 13.99x | **14.96x** |
> | 60  | 9.75x  | 13.99x | **14.97x** |
> | 70  | 11.28x | 13.99x | **15.80x** |
> | 80  | 15.17x | 13.99x | **18.95x** |
> | 90  | 23.97x | 13.99x | **30.49x** |
>
> We show **extended S5.1**: signed-binary achieves the highest arithmetic operation reduction, highest FLOPs, and Highest Speedup (on CPU) for ResNet 18 trained on ImageNet:
>
> *Table: Weight, FLOPs, and Speedup for Different Methods*
>
> | Weight | Binary | Signed Binary w/ Sparsity Support | Ternary w/ Sparsity Support | Signed Binary w/o Sparsity Support | Ternary w/o Sparsity Support |
> |------|------|-------|--------|------|-------|
> | FLOPs (x10^6) (↓) | 282.66 | **238.52** | 422.69 | 279.38 | 450.14 |
> | Speedup (↑) | 1x | **1.26x** | 0.72x | 1.05x | 0.67x |
>
> Further,  we demonstrate efficiency on two devices, S5.1 CPU and S5.2.2 ASIC, and thereby already demonstrate efficiency on general purpose and custom hardware, respectively. We would like to highlight that the GPU kernel developed by Meta [7] is not publicly available. Implementing a novel kernel from scratch is outside the scope of the paper.
>
> [5] Hegde et al. "UCNN: Exploiting computational reuse in deep neural networks via weight repetition." IEEE ISCA 2018.
>
> [6] Prabhakar et al. "Summerge: An efficient algorithm and implementation for weight repetition-aware dnn inference." ACM ICS 2021.
>
> [7] Fu et al. "Q-gym: An equality saturation framework for dnn inference exploiting weight repetition." PACT 2022.

---

### Decision · Action_Editor_XtJJ · 2024-08-26

**Recommendation:** Accept with minor revision

**Comment:**

This paper introduces the repetition-sparsity trade-off concept and PLUM framework for efficient neural network inference. The authors demonstrate PLUM's effectiveness through comprehensive experiments on standard datasets and real hardware.

The authors have addressed most reviewer concerns by adding ablation studies, discussing limitations and practical considerations, and clarifying training details. While some limitations remain, PLUM significantly improves inference efficiency while maintaining accuracy, valuable for resource-constrained environments.

Regarding the minor revision, the AE would like to recommend the authors to
- Add training time cost in the main part of the paper, not in the last discussion session. Provide some numerical results on the time cost;
- Move some contents in appendix I and J to the main part of the paper, as the audience will be interested in seeing larger models;
- Consider adding models larger than ResNet18 for more validation.

**Audience:**

Yes. The audience will be interested in accelerating the inference speed of DNNs.

**Claims And Evidence:**

Yes. This paper introduces the repetition-sparsity trade-off concept and PLUM framework for efficient neural network inference. The authors demonstrate PLUM's effectiveness through comprehensive experiments on standard datasets and real hardware.

The claims are supported by a series of experiments and ablation studies.